# Impact of implementation of the national institute for health and clinical excellence (NICE) head injury guideline in a tertiary care center emergency department: A pre and post-intervention study

**Pratisha Pradhan** [1]☉*, **Alok Pradhan**[1‡], **Anmol Purna Shrestha**[1‡], **Abha Shrestha**[2‡], **Ram Chandra Paudel**[3‡], **Roshana Shrestha**[1]☉

**1** Department of General Practice and Emergency Medicine, Kathmandu University School of Medical Sciences, Dhulikhel, Kavrepalanchok, Bagmati Province, Nepal, **2** Department of Community Medicine, Kathmandu University School of Medical Sciences, Dhulikhel, Kavrepalanchok, Bagmati Province, Nepal, **3** Department of Radiodiagnosis and Imaging, Kathmandu University School of Medical Sciences, Dhulikhel, Kavrepalanchok, Bagmati Province, Nepal

☉ These authors contributed equally to this work.
‡ AP, APS, AS and RCP also contributed equally to this work.
* pratisha.7@gmail.com

## Abstract

### Introduction

Head injury, a common presentation to the emergency department (ED), is a substantial problem in developing countries like Nepal. The current national institute for health and clinical excellence (NICE) guideline revised in January 2014 focuses on effective clinical assessment and early management of head injuries according to their severity in all age groups. This study assessed the impact of implementing this guideline on the proportions of computed tomography (CT) head scans, guideline adherence, and confidence level of the attending physicians.

### Methods

We consecutively recruited 139 traumatic head injury (THI) patients in this prospective pre-post interventional study conducted in the ED of a tertiary care center. We implemented the NICE guideline into routine practice using multimodal intervention through physicians' education sessions, information sheets and guideline-dissemination. The pre and post-implementation CT head scan rates were compared. The post-implementation guideline adherence was assessed. Online Google form-questionnaires including 12 validated case scenarios were distributed to the attending physicians at the end of both phases to assess their confidence levels.

### Results

The implementation resulted in a statistically significant decrease in the proportion of CT head scan rates from 92.0% to 70.0% (p-value = 0.005). Following educational

**Data Availability Statement:** All relevant data are within the paper and its Supporting Information files.

**Funding:** The author(s) received no specific funding for this work.

**Competing interests:** The authors have declared that no competing interests exist.

interventions, improved guideline adherence of 20.3 percentage points (p-value = 0.001) was observed. Nine ED attending physicians were enrolled in the study who showed statistically significant improvement in their confidence level following the intervention. The NICE guideline showed a sensitivity and specificity of 93.6% and 76.4% with 82.6% accuracy compared to that of clinical judgment (100%, 34.6%, and 58.1% respectively) in detecting intracranial lesions.

## Conclusion

The implementation was successful in satisfying the aim of the NICE guideline by decreasing the proportion of CT head scans, improving guideline adherence and increasing the confidence of the attending physicians.

## Introduction

### Background

Traumatic head injury (THI) is a comprehensive term that describes injuries to the scalp, skull and/or underlying tissue in the head due to trauma other than superficial injuries to the face [1]. Worldwide, 69 million individuals are estimated to sustain THI every year with an incidence of mild THI of about 131 cases per 100,000 people, moderate THI of about 15 cases per 100,000 people, and severe THI approximately 14 cases per 100,000 people [2]. Due to the rapid surge in development, motorization and economical liberty, the risk of THI has increased in many Asian countries. Available data show that Asia has the highest percentage of THI due to falls (77.0%), unintentional injuries (57.0%), and road traffic accidents (RTA) (48.0%) [3]. In Nepal, the incidence of THI is estimated to be around 382 per 100,000 with a maximum number of patients presenting with mild THI [4, 5].

Patients with THI initially present to the emergency department (ED) and only a few patients require any intervention. Neurologic examinations and clinical history are a must, followed by a radiological description of THI lesions in imaging. The non-contrast computed tomography (NCCT) has become the investigation of choice for THI cases as it has both sensitivities and specificities approaching 100% for detecting any intracranial lesion [6–8]. In resource-limited settings like ours, CT scans must be ordered thoughtfully due to financial apprehensions. Patients' increased medical expenditures, prolonged ED stays, and concern for adverse effects of radiation exposure and physician's fear of any missed findings and desire to expedite a diagnosis, all affect the decision-making capacity immensely. A composed approach is required to ensure the ordering of head CTs when necessary while vindicating the potential disadvantages of over-imaging. For this, appropriate guidelines can help scrutinize cases [9, 10].

At hospitals in our country, the foremost attending physicians in the ED are the medical officers (MO) who are recent graduates. Their exposure to the ED is limited to about one month during their internship. The decision to order a CT head scan for THI patients is challenging as their experience may not be sufficient in making a rational judgment regarding the use of CT imaging for such patients [9]. Several decision rules [11] are accessible to guide clinical decision-making for the use of CT scans for THI patients and on comparison of these validated rules, the national institute for health and clinical excellence (NICE) criteria showed the highest specificity to identify any intracranial traumatic lesions.

The NICE head injury guideline [1] was developed in 2003 and updated in 2007 that ensured the replacement of skull radiography by CT scans as the prime imaging modality for evaluating THI cases. Then in 2014, the updated NICE guideline focused on earlier imaging and reporting after various systematic reviews. Its sensitivity and specificity for intracranial pathology range from 46.1 to 73.0% and 61% to 82.1% respectively [8, 11].

## Objectives

Despite such multiple validated guidelines and data demonstrating their validity and generalizability, execution of these rules in our clinical settings has been suboptimal. The primary outcome of this study was to assess the impact of implementing the NICE guideline on the proportion of CT scans performed. Furthermore, this study evaluated the guideline adherence and decision-making capacity of the attending physicians, before and after the implementation of the guideline.

## Materials and methods

### Study design

This was a prospective pre-post intervention study.

### Study setting

The study was conducted in the ED of Dhulikhel Hospital-Kathmandu University Hospital (DH-KUH), a community-based teaching hospital serving people from more than 50 out of 75 districts of the country. According to the ED audit of 2018, nearly 20,000 patients visited the ED, of which about 40% were trauma patients that year. It is a 30-bedded ED staffed by 5 faculties, 10–12 MOs and nurses/paramedics. The ED patients are first received by the on-duty triage-health personnel who take a short focused history and vitals to categorize them into trauma or non-trauma and further according to the severity in reference to Rapid Emergency Triage and Treatment System (RETTS) [12]. Following triage, the patient is taken to the respective zones in ED and the treatment is carried out by the attending MO/faculty.

### Study participants

All patients who came to the ED of DH with THI within 24 hours of the incident were enrolled in the study. THI was defined as any injury with impact around the head and/or in the face due to trauma such as RTA, fall injury, physical assault and others (sports injuries, firearm, struck by object and occupation-related injuries). Patients with non-traumatic head injuries i.e. injuries to the brain that are not caused by an external physical force to the head, such as stroke, hypertensive subarachnoid hemorrhage, hypoxic injuries, were excluded from the study. Penetrating head injuries in which the dura mater is breached is an obvious tell-tale-sign of brain injury, and thus were not included in the study. Furthermore, brought dead patients, incomplete data, re-attendees for the same head injury, pregnant women and patient/visitors denying consent were also excluded. For evaluation of self-reported confidence levels of the attending physicians, all the medical officers of the ED participated voluntarily.

### Variables

After ethical approval by the Institutional Review Committee (IRC number 155/18) on December 02, 2018, we developed a proforma for data collection which was validated by all five faculty members of the ED. Necessary amendments were made and finalized after the pilot testing in ten THI patients not included in the study.

The attending physician recorded data in the predesigned proforma which included hospital identification number, demographic data {age (<16 years and ≥16 years), gender, (male or female)}, triage category (red, orange, yellow, green), mechanism of the injury (RTA, fall, assault, others), mode of transport (self or ambulance), any pre-hospital treatment, the presence of influence of drugs/alcohol and any comorbidities. Glasgow coma scale (GCS) [13] on arrival (13–15, 9–12, ≤ 8 as mild, moderate and severe head injuries, respectively), physician's clinical diagnosis and information on whether CT was sent or not were noted. If CT was sent, the indication of CT head according to the physician's opinion and CT head findings as stated by a radiologist were recorded. Lastly, the disposition of the patients from the ED (discharge, admission, referral and mortality) was mentioned. The outcome of each patient who was discharged from the ED without a CT head or who refused the investigation was followed up on the 10th day and his/her health status was inquired and documented.

An online Google form–questionnaire was sent via email to all the attending physicians that contained 12 case scenarios (four pediatric and eight adults) related to trauma for which they had to decide whether or not head CT was indicated. If they assumed indicated, they were asked to mention the indication for CT according to their opinion. Furthermore, the level of confidence using a Likert scale of 10 (1 = low and 10 = high) was used to determine their confidence level to order a head CT for each case. The 12 case scenarios were referenced from real cases visiting ED of DH and were validated by the ED faculties.

## Study procedure

**Pre-implementation period.** The data collection was carried out from 2nd February, 2019 to 28th March, 2019, only after the initial stabilization of the patient and hence patient management was not delayed by the research. Informed written consent was taken from the patient or the patient's key caretaker (in case the patient's condition was not sound enough to give consent) meeting the inclusion criteria. The indication of the CT head was decided by the physicians according to their clinical judgment. We reviewed the electronic medical records of the samples to determine if the CT had been requested in reference to the NICE guideline. The online Google-questionnaire to assess the knowledge and confidence was sent to all the attending physicians of the ED and their responses were collected.

**Intervention period.** After completion of the pre-implementation period, a multimodal intervention was carried out to implement the NICE guideline into routine practice at ED. Components of the intervention included the clinical endorsement of the guideline in the ED, staff education and guideline dissemination. A didactic case-based teaching session was conducted for all the clinical staff of ED including faculties, resident doctors, attending physicians, nurses and paramedics. During the session, the current departmental pattern regarding head CTs and results of the pre-implementation phase was shared, followed by an explanation of the flowchart depicting the NICE guideline to enhance their clinical decision-making capability. Posters illustrating the NICE guideline flowchart were displayed in all zones of ED.

**Post-implementation period.** In the post-implementation period (1st May, 2019 to 20th June, 2019), a checklist of the NICE guideline CT head indications was attached along with the proforma. The attending physicians of the ED collected the data in a similar fashion. We reinforced the adherence to the guideline by working on-site for the first week of its implementation. We collected such duly filled proforma from the ED and the electronic medical record search corroborated the patient information.

After the post-implementation phase, the online Google form-questionnaire containing the same 12 case scenarios were distributed to the attending physicians and their confidence level was explored.

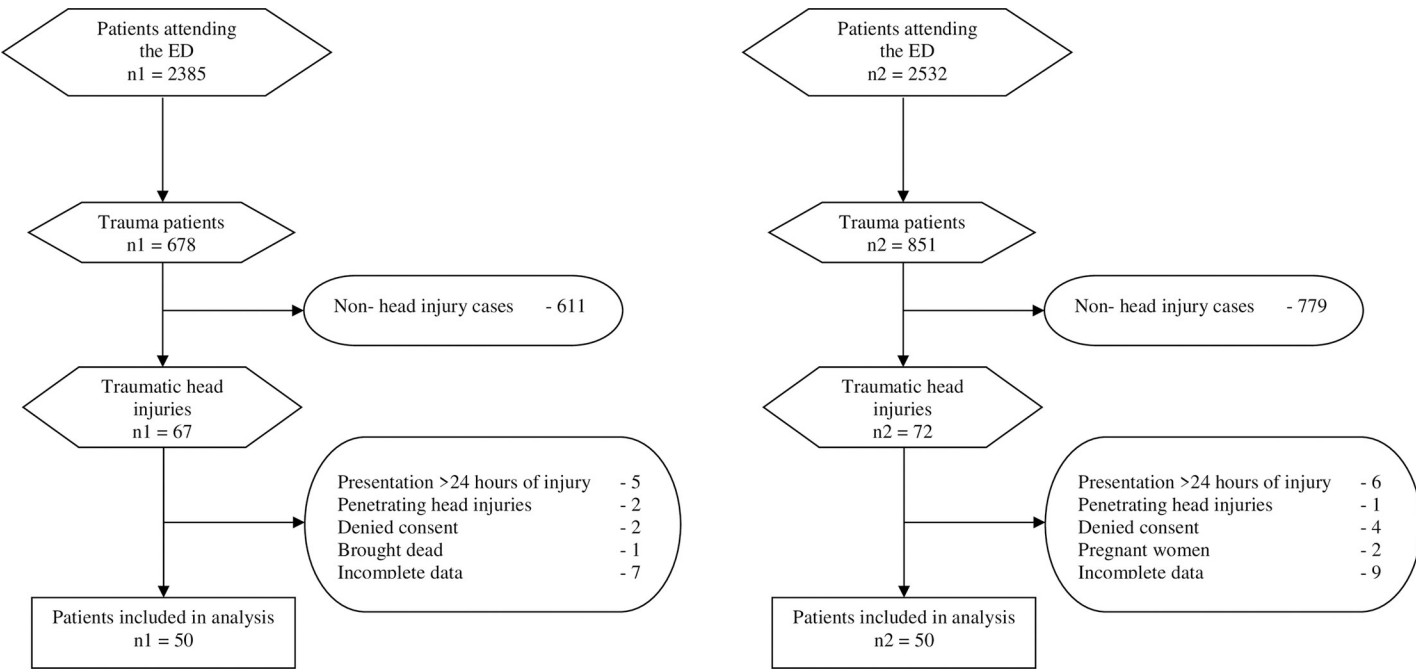

**Fig 1. Standards for Reporting Diagnostic Accuracy studies (STARD) [15] flow diagrams for patients who presented with THI in pre and post-implementation periods respectively.**

## Study size

This study considered a 95% confidence interval, 80% power, an equal-intervention/control ratio, margin of error of 0.3 and a drop rate of 25%. Since the proportions of CT head scans performed in the pre and post-implementation of the NICE guideline were not known, we considered it to be 50%. Using the standard statistical formula, the sample size to compare two proportions was calculated to be 46 in each group. Thus, consecutive sampling of 50 patients was done in each study period [14] as shown in Fig 1.

## Statistical methods

The data from the patients were entered into a Microsoft Excel spreadsheet and the data from the Google forms were downloaded in the excel spreadsheet. The data analysis was done using Statistical Package for the Social Sciences (SPSS) version 21.

For descriptive statistics, frequency (percentage) and mean with standard deviation (SD) or median with interquartile range (IQR) were used for continuous parameters whichever applicable. Graphical and tabular presentations were also plotted. The categorical variables were analyzed by cross-tabulation using chi-square. Due to a small sample group and asymmetrically distributed data, the t-test was complemented with a Mann-Whitney U-test to compare the medians of the continuous independent variables. Diagnostic tests were estimated using 2*2 tables. The performance of the NICE guideline was calculated using sensitivity, specificity, positive predictive value (PPV), negative predictive value (NPV) and accuracy. The scans sent on the patient's request were excluded for comparison of the number of CT scans performed with and without indication and guideline adherence in the pre and post-implementation periods. The pre-and the post-implementation Likert scale ratings for the confidence level of the attending physicians were expressed as medians with IQR and compared with Wilcoxon signed-rank test.

## Results

Following the implementation of the NICE guideline in this study, there was a decrease in the proportion of CT head scans performed, increased guideline adherence and an increase in the confidence levels of the attending physicians to indicate a CT head scan in cases of THI. The proportion of CT head scans performed decreased from 46 (92.0%) to 35 (70.0%) in our study. There was also a decrease in the proportion of CT head scans performed without indication from 17 (42.5%) to 6 (22.2%) and improved adherence in relation to CT scanning of the head from 57.5% to 77.8%. A total of nine ED attending physicians were enrolled in the study who showed statistically significant improvement in their confidence level post-intervention.

### Baseline characteristics

The demography, incident details and clinical parameters of the study population are illustrated in Table 1. A comparison of the baseline characteristics of both the study populations was made which showed that they were similar groups of samples.

### Performance of the NICE guideline

In total, 81 CT head scans were performed during the study period of which 14 (six and eighteen for pre-and post-implementation phases respectively) were performed on patient request. On excluding those, in the pre-implementation period, 40 out of the 44 THI patients required a head CT scan on retrospective implementation of the guideline. In the post-implementation period, 27 out of 42 THI patients required a head CT scan according to the NICE guideline. The THI patients who did not undergo a CT head were followed up through phone calls and none had any danger signs or required re-visits. The diagnostic evaluation of the NICE guideline in comparison to the physician's clinical judgment was done using a 2*2 table and calculated as shown in Tables 2 and 3, with the CT findings and final outcome kept as the gold standard. A 100% guideline adherence was not achieved in the post-implementation period in our study.

### Proportion of CT head scans and CT findings in the pre and post-implementation periods

The proportion of CT head scans performed in the pre and the post-implementation periods are shown in Table 4. In comparison, there was a significant decrease in the proportion of CT scans performed after the implementation of the guideline (p-value = 0.005). There were however 2 cases in the pre-implementation period who underwent CT head scan without indication as per NICE guideline and the report was abnormal.

### Guideline adherence

If the NICE guideline were to be applied on the pre-implementation phase–study population, of the total 46 CT head scans performed, 50.0% were indicated while 50.0% were not indicated. Likewise in the post-implementation phase, of the 35 CT head scans performed, 60.0% were indicated while 40.0% were not indicated. Following exclusion of the CT head scans performed on patient request, the proportion of the CTs performed without indication decreased by 20.3 percentage points (from 42.5%-22.2%), while the proportion of indicated CT scans increased by 20.3 percentage points (from 57.5%-77.8%) which is a statistically significant improved adherence (chi-squared p = 0.001) following the implementation of the NICE guideline. (Table 5)

**Table 1. Demography, incident details and clinical parameters of the study population, n = 100.**

| S. No | Baseline Characteristics | Pre implementation of NICE N1 = 50 | Post implementation of NICE N2 = 50 | p- value |
|---|---|---|---|---|
| 1 | Age Category (years) n (%) | | | 0.334[a] |
| | ≤ 15 | 10 (20.0) | 13 (26.0) | |
| | 16–64 | 35 (70.0) | 30 (60.0) | |
| | > 65 | 5 (10.0) | 7 (14.0) | |
| 2 | Gender n (%) | | | 0.296[a] |
| | Female | 10 (20.0) | 17 (34.0) | |
| 3 | Triage Category n (%) | | | 0.843[a] |
| | Red | 22 (44.0) | 6 (12.0.0) | |
| | Orange | 8 (16.0) | 12 (24.0) | |
| | Yellow | 20 (40.0) | 27 (54.0) | |
| | Green | 0 (0) | 5 (10.0) | |
| 4 | GCS on Arrival | | | 0.059[b] |
| | Median (IQR) | 15 (14.75–15.00) | 15 (15.00–15.00) | |
| | Mild | 42 (84.0) | 46 (92.0) | |
| | Moderate | 2 (4.0) | 3 (6.0) | |
| | Severe | 6 (12.0) | 1 (2.0) | |
| 5 | Mechanism of Injury n (%) | | | 0.379[a] |
| | Fall | 24 (48.0) | 23 (46.0) | |
| | Physical Assault | 10 (20.0) | 7 (14.0) | |
| | RTA | 13 (26.0) | 14 (28.0) | |
| | Others | 3 (6.0) | 6 (12.0) | |
| 6 | Mode of Transport n (%) | | | |
| | Self | 26 (52.0) | 33 (66.0) | 0.197[a] |
| | Ambulance | 24 (48.0) | 17 (34.0) | |
| 7 | Taken pre- hospital treatment n (%) | 2 (4.0) | 6 (12.0) | 0.594[a] |
| 8 | Under influence of Alcohol/ Drugs n (%) | 4 (8.0) | 6 (12.0) | 0.441[a] |
| 9 | Presence of Comorbidities n (%) | 2 (4.0) | 2 (4.0) | 0.768[a] |
| 10 | Disposition from the ED n (%) | | | 0.088[a] |
| | Discharge | 20 (40.0) | 26 (52.0) | |
| | Admission | 14 (28.0) | 5 (10.0) | |
| | Referral | 5 (10.0) | 4 (8.0) | |
| | DOR/LAMA | 11 (22.0) | 15 (30.0) | |

[a]-Chi-square

[b]-Mann-Whitney U test; NICE-National institute for health and clinical excellence; IQR-Interquartile range; GCS-Glasgow coma scale; DOR-Discharge on request; LAMA-Leave against medical advice.

**Table 2. Performance of the NICE guideline in comparison to the clinical judgment, n = 86.**

| | Clinical judgment N = 86 | | NICE guideline N = 86 | |
|---|---|---|---|---|
| | CT head scan not indicated | CT head scan indicated | CT head scan not indicated | CT head scan indicated |
| THI n (%) | 0 (0) | 31 (36.1) | 2 (2.3) | 29 (33.7) |
| No THI n (%) | 19 (22.1) | 36 (41.8) | 42 (48.8) | 13 (15.2) |

NICE-National institute for health and clinical excellence; CT-Computed tomography; THI-Traumatic head injury (as indicated by CT head finding and outcome of the patient).

**Table 3. Diagnostic tests.**

| CT Indication | Sensitivity | Specificity | PPV | NPV | Accuracy |
|---|---|---|---|---|---|
| Clinical judgement % (95% CI) | 100 (88.8–100.0) | 34.6 (22.2–48.6) | 46.3 (41.6–51.1) | 100 (82.4–100.0) | 58.1 (47.0–68.7) |
| NICE % (95% CI) | 93.6 (78.6–99.2) | 76.4 (62.9–86.8) | 69.1 (57.9–78.4) | 95.5 (84.5–98.8) | 82.6 (72.9–89.9) |

CI-Confidence Interval; NICE-National institute for health and clinical excellence; PPV-Positive predictive value; NPV-Negative predictive value.

**Table 4. Proportion of CT head scans and CT findings in the pre and post-implementation phases of the study, n = 100.**

| | Pre-implementation of NICE N1 = 50 | Post-implementation of NICE N2 = 50 |
|---|---|---|
| CT Performed n (%) | 46 (92.0) | 35 (70.0) |
| Normal | 31 (67.4) | 19 (54.3) |
| Abnormal | 15 (32.6) | 16 (45.7) |
| Fractures | 14 (93.3) | 9 (52.9) |
| With pneumocephalus | 1 (6.7) | 3 (17.7) |
| Intracranial hemorrhages | 10 (66.7) | 10 (58.8) |
| (EDH/ SDH/ SAH/ IPH) | | |

NICE-National institute for health and clinical excellence; CT-Computed tomography; EDH-Extradural hematoma; SDH-Subdural hematoma; SAH-Subarachnoid hemorrhage; IPH-Intraparenchymal hemorrhage.

**Table 5. Guideline adherence during different phases, n = 86.**

| CT performed/not performed according to Attending Physician's Clinical Judgement | Pre-implementation of NICE N1 = 44 | | Post-implementation of NICE N2 = 42 | |
|---|---|---|---|---|
| | CT Head Scan not indicated | CT head scan indicated | CT Head Scan not indicated | CT head scan indicated |
| CT head not performed n = 19 n (%) | 4 (100) | 0 (0) | 15 (100) | 0 (0) |
| CT head performed n = 67 n (%) | 17 (42.5) | 23 (57.5) | 6 (22.2) | 21 (77.8) |

NICE-National institute for health and clinical excellence; CT-Computed tomography.

## Evaluation of the ED attending physicians

The median scores of the questionnaire were 11 in both pre-implementation (IQR: 9.0–11.5) and post-implementation (IQR: 11.0–12.0) phases. However, the score ranged from 8 to 12 and 10 to 12 in the pre and post-implementation periods respectively. The scores were compared using the Wilcoxon signed-rank test that showed a significant difference ($p$-value = 0.047) as shown in Fig 2.

The level of confidence to indicate or not indicate the CT scan was evaluated using a Likert scale of 10 for each case. Table 6 shows that 9 out of 12 cases showed statistically significant differences in terms of improvement in the confidence level. In 3 case scenarios, there was no statistically significant improvement.

## Discussion

The importance of the problem of THI is always underestimated due to the lack of research and good quality data. This is a study in Nepal to investigate the impact and adherence of a head injury guideline at DH-ED. Following the implementation of the NICE guideline in this

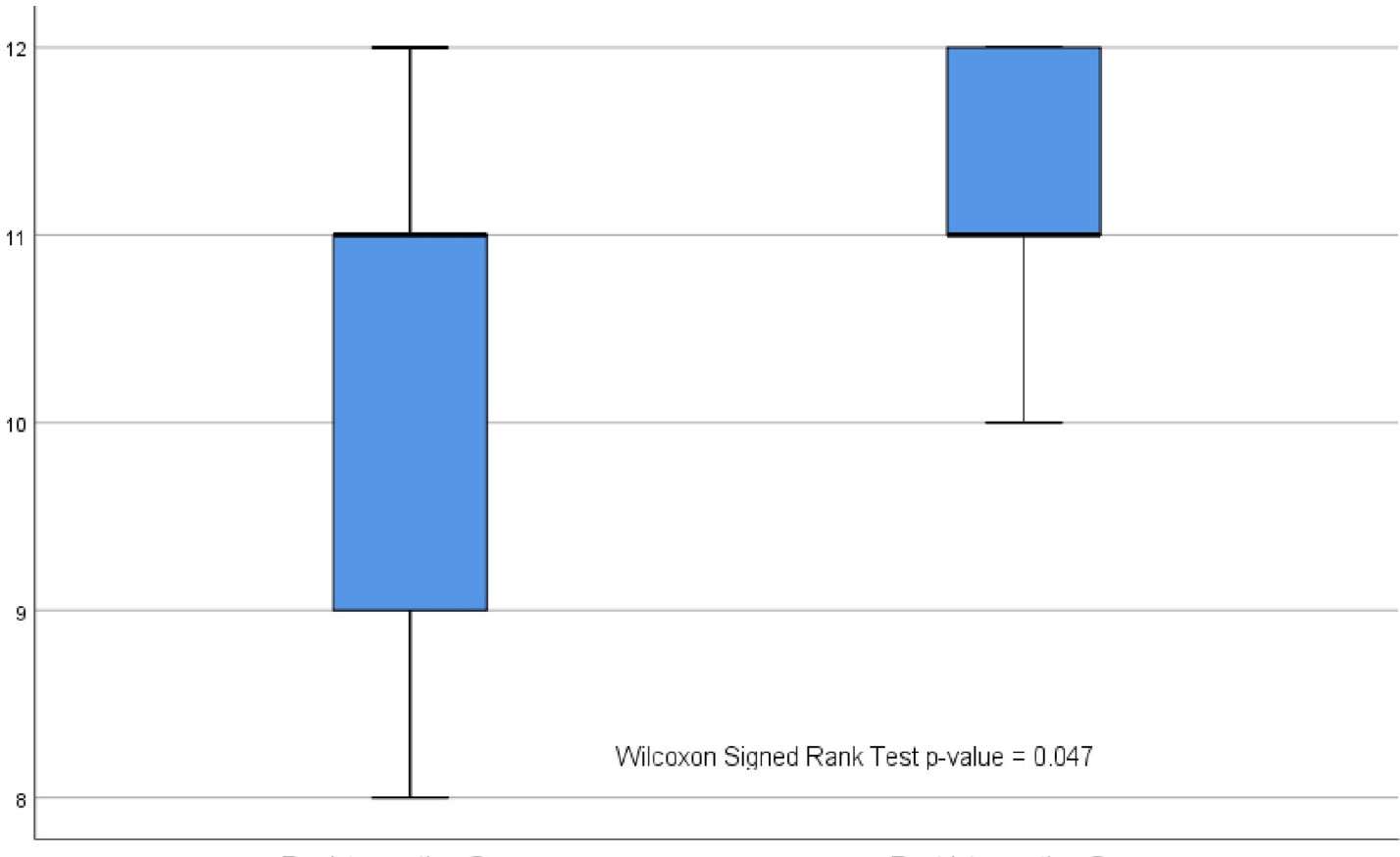

**Fig 2. Box plot showing pre and post-implementation scores.**

study, there was a decreased proportion of CT head scans performed, increased guideline adherence and an increase in the confidence levels of the MOs to indicate a CT head scan in cases of THI.

**Table 6. Confidence of the ED attending physicians in relation to the 12 case scenarios.**

| Cases | Pre-implementation confidence Median (IQR) | Post-implementation confidence Median (IQR) | p-value[a] |
|---|---|---|---|
| 1 | 8 (5.0–9.5) | 10 (10–10) | **0.018** |
| 2 | 8 (4.0–8.0) | 10 (9–10) | **0.049** |
| 3 | 8 (4.5–9.0) | 10 (9.5–10.0) | **0.017** |
| 4 | 7 (4.0–7.0) | 9 (7.5–10.0) | **0.029** |
| 5 | 8 (4.5–8.5) | 10 (10.0–10.0) | **0.007** |
| 6 | 7 (5.5–8.5) | 10 (9.5–10.0) | **0.007** |
| 7 | 7 (6.0–9.0) | 10 (7.0–10.0) | **0.048** |
| 8 | 9 (6.0–10.0) | 10 (9.5–10.0) | 0.129 |
| 9 | 9 (5.5–10.0) | 10 (9.5–10.0) | 0.174 |
| 10 | 9 (5.5–10.0) | 10 (9.5–10.0) | 0.102 |
| 11 | 9 (7.0–10.0) | 10 (10.0–10.0) | **0.038** |
| 12 | 8 (4.5–9.0) | 10 (10.0–10.0) | **0.011** |

[a]Wilcoxon signed-rank test; IQR-Interquartile range.

## Demography

This study shows that the median age group sustaining head injuries was 27 years (IQR: 16.0–42.5). The most common age group vulnerable for injury in this study was 16–65 years, followed by <16 years, ≥65years (65.0%, 23.0% and 12.0% respectively) which is similar to the findings of various studies [16–19]. A reason for this could be that the exposure of young people to accidents in traffic and unsafe working situations. The average life expectancy in Nepal is rather low (67.5 years) and the median age is 23.4 years which would explain the minimal number of old age people sustaining THI [20].

A significant gender disparity is noted in this study, with male (73.0%) being more prone to any injury. The preponderance of males over females is noted in all modalities of injuries in our study, as in many studies [16–23] conducted all around the world.

## Performance of the NICE guideline

Compared to the traditional clinical judgment, the NICE guideline showed a sensitivity and specificity of 93.6% and 76.4% with 82.6% accuracy in this study. Had the adherence been better, the post-implementation period guideline specificity and PPV would have increased more. These values were similar to the systematic review [24] done in Italy. Lower sensitivities (82.1% and 72.5%) and specificities (46.1% and 60.9%) were shown in multicenter validation studies [8, 11] in Netherland and concluded that the lowest number of patients requiring scan for either of the outcomes was reached with the NICE criteria. In an audit in England and Wales [21], the NICE guideline resulted in 100% sensitivity and 93.8% specificity which was quite high compared to our results.

In our study, there were two positive findings in the non-indicated CT scans during the pre-implementation period. The first case was a 24 years male who presented with a history of RTA sustaining injury mostly on the face. He was under the influence of alcohol and thus CT head scan was ordered as per the clinical judgment. The CT finding was a contusion in the right frontal and parieto-occipital region with a depressed comminuted fracture in the frontal bone with orbital extensions, fractured nasal bones with overlying soft tissue swelling/hematoma and orbital emphysema. The NICE guideline makes no recommendations for THI patients with a history of alcohol consumption in contrast to the New Orleans Criteria (NOC) which recommends head CT scans for THI patients under any drug or alcohol influence [25].

The next case was 12 years, male with a history of fall from a height of about two meters, sustaining an injury on the right side of the head. Although there was no indication for CT head according to the NICE guideline in this case, when it comes to children, it is very difficult to decide. Thus the clinicians mostly interpret the criteria according to the clinical situation faced by them in the ED.

## Proportion of CT head scans performed

The proportion of CT head scans performed decreased from 46 (92.0%) to 35 (70.0%) in our study. The result is contradictory in various previous studies [21]. Correspondingly there have been studies [26, 27] showing a similar decrement in the proportion of CT head scans performed following the implementation of the NICE guideline. A previous study [22] showed that the proportion of CT head scans performed decreased by a lesser rate than our study, from 40 (4.8%) to 23 (2.4%) and the proportion of CT head scans performed without indication decreased from 14 (1.7%) to 4 (0.4%). On the contrary, the implementation of the NICE guideline in the ED of a teaching hospital and a District General Hospital led to a two to five-fold increase in the CT head scan rate [16].

Following interventions for quality improvement to adhere to the NICE guideline in a study [27], there was a statistically significant decrease of 23.0 percentage points in the number of CT heads requested with no clear indication following intervention (p = 0.00027) which was similar to our results (20.3%, p = 0.001). This reduction in unnecessary CT head scans may result in considerable savings to numerous families while ensuring access for those really in need. In addition, improved triage of head injuries in the rural communities would help in needful referral for further imaging and earlier access to care. These changes could potentially reduce costs involved with transport, CT scanning, and also reduce the strain on emergency, neurology, and radiology departments.

### Guideline adherence

We found that the implementation of the NICE guideline was associated with improved adherence in relation to CT scan of the head from 57.5% to 77.8%. This improvement of 20.3% represents both a reduction in unnecessary CT usage and thus radiation exposure. This increment in adherence in our study could be explained by the multiple approaches: teachings during the implementation phase, posters of the guideline to help remind all the attending physicians of the indications for CT scans. We reinforced the adherence to the NICE guideline by supervising the MOs on-site in the first week of implementation.

Similar findings were noted in the study conducted at a major children's hospital ED which showed a significant increase in guideline adherence from 79.2% to 85.1% [22]. The study by Mooney et al. [26] showed compliance of 94.2% pre-implementation of NICE guideline which increased to 98.8% post-guideline implementation in adults. The studies by Harris et al. [27] and Dhungana et al. [17] also concluded that the adherence to the NICE guideline could be improved by interventions such as an educational program to the trauma team of the hospital, a repeated survey of the attending physicians and their feedback.

### Evaluation of the ED attending physicians

The decision to order a head CT for patients with THI is quite challenging, particularly in mild THIs. The actual degree of harm (radiation exposure, false-positive findings) or benefit (discovery of treatable intracranial injury) is not comprehensively known. Nonetheless, the attending physicians must make this decision in almost every shift. Many clinical and non-clinical factors influence their decision to order a head CT [9]. The current research focused on their confidence level to make the decision with and without a guideline protocol. There was statistically significant increased self-reported confidence after the guideline implementation.

In three case scenarios, there was no statistically significant improvement in their level of confidence. The cases were a child with a dangerous mechanism of injury with four episodes of vomiting, an adult with one post-traumatic seizure and another adult with GCS <13 at presentation. Since the indications for head CT were quite obvious, the MOs were confident to indicate the CT scan even before the implementation.

### Limitations

These before and after type of studies have the strength of suggesting that the outcome is impacted by the intervention, however there are a number of limitations. Firstly, we did not have control over elements that are changing at the time of intervention such as joining of a new attending physician or start of neurosurgical department. Although, the pre and post-implementation evaluation of confidence levels of the attending physicians were done among the same set and no new physicians were included in between, this study did not take into account the likelihood of prior knowledge of the attending physicians about the NICE

guideline or any other head injury guidelines prior to the intervention. Furthermore, convenient sampling method used in this study causes sampling bias and some sample have been missed in between. Randomized control trial (RCT) would have been a better study design but randomization was not feasible in our setting. Guideline adherence with respect to the timeline was not focused in this study and the reporting of the CT head scans done at off-hours were not available within 1 hour of the investigation. However, it was available by the first working hour of the next day. Even in patients with normal head CT results and no identified intracranial injuries, there may have been subtle brain injury which would have been better detected by an MRI [28] or perhaps a second CT scanning, neither of which were obtained in such patients.

## Conclusion

The overall compliance with the NICE guideline was achieved and there was a reduction in unnecessary CT scans and thus exposure to radiation. No patient with THI was missed with the implementation of the guideline and the attending physicians felt more confident to decide on indications for CT head scans for THI cases after the implementation of the NICE guideline.

Furthermore, strategies leading to protocol-guided head CT ordering reduce the rates of CT scans and thus reduce the costs. This leaves another space for research regarding the cost-benefit as an impact of the implementation of the NICE guideline in a low-middle income country like Nepal.

## Supporting information

**S1 File. Pre-implementation questionnaire—Google form.**
(PDF)

**S2 File. Post-implementation questionnaire—Google form.**
(PDF)

**S3 File. Implementation of NICE guideline in ER.**
(PDF)

**S4 File. Flowchart posters for ER.**
(PDF)

## Acknowledgments

Medical officers (Anuradha Pradhan, Sareen Shrestha, Manoj Dawadi, Suyog Shrestha, Hema Joshi, Prem Waiba, Rahul Neupane, Ambika Belbase and Vishnu Khadka) of the Department of General Practice and Emergency Medicine, DH, for assisting in data collection. ED faculties for their expert opinions and guidance. All the participants for their enthusiastic participation.

## Author Contributions

**Conceptualization:** Pratisha Pradhan, Roshana Shrestha.

**Data curation:** Roshana Shrestha.

**Formal analysis:** Pratisha Pradhan, Anmol Purna Shrestha, Abha Shrestha, Roshana Shrestha.

**Investigation:** Pratisha Pradhan, Ram Chandra Paudel.

**Methodology:** Pratisha Pradhan, Alok Pradhan, Anmol Purna Shrestha, Abha Shrestha, Roshana Shrestha.

**Software:** Alok Pradhan, Abha Shrestha, Roshana Shrestha.

**Supervision:** Anmol Purna Shrestha, Roshana Shrestha.

**Writing – original draft:** Pratisha Pradhan.

**Writing – review & editing:** Pratisha Pradhan, Alok Pradhan, Anmol Purna Shrestha, Abha Shrestha, Ram Chandra Paudel, Roshana Shrestha.

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
