## [Decision Letter · Decision Letter 0]

5 Jun 2021

PONE-D-21-09113

Impact of implementation of the national institute for health and clinical excellence (NICE) head injury guideline in a tertiary care center emergency department: A pre and post-intervention study.

PLOS ONE

Dear Dr. Pradhan,

Thank you for submitting your manuscript to PLOS ONE. After careful consideration, we feel that it has merit but does not fully meet PLOS ONE’s publication criteria as it currently stands. Therefore, we invite you to submit a revised version of the manuscript that addresses the points raised during the review process.

Both reviewers show support for this manuscript although one reviewer has raised concerns about the presentation of the data that will need to be addressed in a revised version.  Additionally, one reviewer also raises the limitations of before and after studies and this is an important point.  The manuscript should include acknowledgement of the challenges and limitations of before and after studies, and the potential for bias in the findings. Discussion of why an RCT was not performed, and justification for the before and after approach would be beneficial.

We look forward to receiving your revised manuscript.

Kind regards,

Belinda J Gabbe, PhD

Academic Editor

PLOS ONE

Journal Requirements:

2. Thank you for stating in the text of your manuscript "Informed written consent was taken from the patient or the patient’s key caretaker (in case the patient’s condition was not sound enough to give consent) meeting the inclusion criteria". Please also add this information to your ethics statement in the online submission form.

3. Please provide the date range of the pre-implementation and post-implementation periods.

4. Please provide copies of all materials used in the teaching sessions and the posters that were posted as suplementary files.

5. Please ensure that you refer to Figure 1 in your text as, if accepted, production will need this reference to link the reader to the figure.

Reviewers' comments:

Reviewer's Responses to Questions

**Comments to the Author**

1. Is the manuscript technically sound, and do the data support the conclusions?

Reviewer #1: Partly

Reviewer #2: Yes

2. Has the statistical analysis been performed appropriately and rigorously? 

Reviewer #1: Yes

Reviewer #2: Yes

3. Have the authors made all data underlying the findings in their manuscript fully available?

Reviewer #1: Yes

Reviewer #2: Yes

4. Is the manuscript presented in an intelligible fashion and written in standard English?

Reviewer #1: Yes

Reviewer #2: Yes

5. Review Comments to the Author

Reviewer #1: I would like to thank the authors for this study which looked at the impact of NICE guidelines on head injuries using a pre and post intervention analysis. A significant decrease in the number of CT and a significant guideline adherence were observed post implementation. It is well known that pre-post study designs can lead to erroneous results in the absence of a control group. This is my main concern in regards to this study.

I have some comments :

- lines 259, 260 and 342 the difference in percentage is not expressed as percentage but in number of percentage point.

Instead of "..decreased by 20.3% ....... increased by 20.3% ..." should read " "..decreased by 20.3 percentage points ....... increased by 20.3 per. For rate of increase use (new-old)/old*100.

-Line 333 : Even in the reference [27] they made the same mistake.

-Table 3: Some miscalculations in Table 3:

For Clinical judgement For NICE

PPV 46.3% (82.4% - 100%) PPV 69% (52.9% - 82.4%)

NPV 100% (82.4% 0 100%) NPV 95.5% (84.5% - 99.4%)

Acc 58.1% (47.7% - 68.5%) Acc 82.6% (74.6% - 90.6%)

Reviewer #2: Thank you for submitting this paper for publication. Although the numbers of patients with head injury are low, the results of the study are significant in that the introduction of the NICE Head Injury Guidelines resulted in more appropriate ordering of CT Brain scans in patients with head injuries.

6. PLOS authors have the option to publish the peer review history of their article (what does this mean?). If published, this will include your full peer review and any attached files.

Reviewer #1: No

Reviewer #2: No

---

## [Author Response · Author response to Decision Letter 0]

8 Jun 2021

Revisions made are as follows:

Response: Thank you for the suggestion. According to the formatting sample provided,

In line 23, “Introduction/Study Objectives” has been replaced with “Introduction”.

In line 138 “Supplement 1 and 2” have been deleted.

In line 232 – Inside the table

“CT Head Scan not indicated” has been replaced with “CT head scan not indicated”

A line 263 has been added and subsequently line numbers have changed accordingly.

In line 299 “studies [16–23]conducted” a space has been added after the reference and now it is corrected to “studies [16–23] conducted”

In line 356, “Evaluation of the ED Attending Physician” has been replaced with “Evaluation of the ED attending physicians”

One of the reviewers had also raised the acknowledgement of the challenges and limitations of before and after studies, and the potential for bias in the findings, why an RCT was not performed, and justification for the before and after approach would be beneficial. Hence, these suggestions have been duly noted and have been added.

From lines 370 to 385, corrections were made to the limitations of the study:

“These before and after type of studies have the strength of suggesting that the outcome is impacted by the intervention, however there are a number of limitations. Firstly, we did not have control over elements that are changing at the time of intervention such as joining of a new attending physician or start of neurosurgical department. Although, the pre and post-implementation evaluation of confidence levels of the attending physicians were done among the same set and no new physicians were included in between, this study did not take into account the likelihood of prior knowledge of the attending physicians about the NICE guideline or any other head injury guidelines prior to the intervention. Furthermore, convenient sampling method used in this study causes sampling bias and some sample have been missed in between. Randomized control trial (RCT) would have been a better study design but randomization was not feasible in our setting. Guideline adherence with respect to the timeline was not focused in this study and the reporting of the CT head scans done at off-hours were not available within 1 hour of the investigation. However, it was available by the first working hour of the next day. Even in patients with normal head CT results and no identified intracranial injuries, there may have been subtle brain injury which would have been better detected by an MRI [28] or perhaps a second CT scanning, neither of which were obtained in such patients.”

Line 499: “Supporting Information” has been replaced with “Supporting information”

The following file names have been corrected as per the style requirements.

Line 500 S1 File. Pre-implementation questionnaire - Google forms.

Line 501 S2 File. Post-implementation questionnaire - Google forms.

Their respective file names have been renamed and submitted.

Also, lines 504 to 512 have been deleted as they were not required in the supporting information section. Rather, the figures have been uploaded to the Preflight Analysis and Conversion Engine (PACE) digital diagnostic tool and submitted.

2. Thank you for stating in the text of your manuscript "Informed written consent was taken from the patient or the patient’s key caretaker (in case the patient’s condition was not sound enough to give consent) meeting the inclusion criteria". Please also add this information to your ethics statement in the online submission form.

Response: Thank you for the acknowledgement of the statement. This information has been added to the ethics statement in the online submission form.

3. Please provide the date range of the pre-implementation and post-implementation periods.

Response: Thank you for the suggestion. The date range of pre and post-implementation periods have been added.

In line 147, “from 2nd February, 2019 to 28th March, 2019,” has been added.

In line 166, “(1st May, 2019 to 20th June, 2019)” has been added.

4. Please provide copies of all materials used in the teaching sessions and the posters that were posted as supplementary files.

Response:

The respective files have been submitted which were used in the teaching sessions and poster illustrations.

Line 502 S3 File. Implementation of NICE guideline in ER.

Line 503 S4 File. Flowchart posters for ER.

5. Please ensure that you refer to Figure 1 in your text as, if accepted, production will need this reference to link the reader to the figure.

Response: Thank you for the correction. Figure 1 has been referred in the text. 

In line 180, “as shown in Figure 1” has been added.

Response: The references have been rechecked.

Line 443 – “Available from: https://doi.org/10.1007/BF01405862” has been added.

Line 454 Reference no. 17. “Dhungana S, Shrestha MK, Ghartimagar D, Ghosh A. Emergency Imaging of Head and Cranio-Facial Injuries:Implementing NICE Guidelines-A Cross Sectional Analysis from Western Region of Nepal. CMRA [Internet]. 2015 [cited 2020 Jun 17];3:218. Available from: http://www.pubmedhouse.com/journals/ms/articles/PMHID1047abstract.html”

Replaced with

“Dhungana S, Shrestha MK, Ghartimagar D, Ghosh A. Emergency imaging of head and cranio-facial injuries: Implementing NICE guidelines – a cross sectional analysis from western region of Nepal. ms [Internet]. 2015 [cited 2021 Jun 8];3:218–24. Available from: https://cmrasociety.org/journal/index.php/ms/article/view/54”

Line 483 – Reference 25. “Kavalci C, Aksel G, Salt O, Yılmaz M, Demir A, Kavalci G, et al. Comparison of the Canadian CT head rule and the New Orleans criteria in patients with minor head injury. World journal of emergency surgery : WJES. 2014;9:31.” 

Has been replaced with:

“Kavalci C, Aksel G, Salt O, Yilmaz MS, Demir A, Kavalci G, et al. Comparison of the Canadian CT head rule and the new orleans criteria in patients with minor head injury. World J Emerg Surg [Internet]. 2014 [cited 2021 Jun 8];9:31. Available from: https://www.ncbi.nlm.nih.gov/pmc/articles/PMC3997198/”

7. Review Comments to the Author

Reviewer #1: I have some comments:

- lines 259, 260 and 342 the difference in percentage is not expressed as percentage but in number of percentage point.

Instead of "..decreased by 20.3% ....... increased by 20.3% ..." should read " "..decreased by 20.3 percentage points ....... increased by 20.3 per. For rate of increase use (new-old)/old*100.

-Line 333 : Even in the reference [27] they made the same mistake.

Response: Thank you for pointing out the mistakes. The corrections have been made as suggested.

Line 41 – “guideline adherence of 20.30%” was replaced with “guideline adherence of 20.3 percentage points”

In lines 259, 260 and 261,

Instead of "decreased by 20.3%" it has been replaced with "decreased by 20.3 percentage points” and instead of “increased by 20.3%” it has been replaced with “increased by 20.3 percentage points”

In line 334, similar corrections have been made,

“decrease of 23.0%” has been replaced with “decrease of 23.0 percentage points”

-Table 3: Some miscalculations in Table 3:

For Clinical judgement For NICE

PPV 46.3% (82.4% - 100%) PPV 69% (52.9% - 82.4%)

NPV 100% (82.4% - 100%) NPV 95.5% (84.5% - 99.4%)

Acc 58.1% (47.7% - 68.5%) Acc 82.6% (74.6% - 90.6%)

Response: 

Regarding line 236 Table 3, I would like to clarify and correct my data results:

With prevalence of 36.05%, I had used the formulae as follows, 

PPV = (Sensitivity * Prevalence)/ [(Sensitivity * Prevalence) + ((1 - Specificity) * (1 - Prevalence))]

 NPV = (Specificity * (1 - Prevalence))/ [((1 - Sensitivity) * Prevalence) + (Specificity * (1 - Prevalence))]

Accuracy = Sensitivity × Prevalence + Specificity × (1 − Prevalence)

Confidence intervals for sensitivity, specificity and accuracy are "exact" Clopper-Pearson confidence intervals.

Confidence intervals for the predictive values are the standard logit confidence intervals given by Mercaldo et al. 2007.

The corrected table is as follows:

Line 236 - Table 3: Diagnostic tests.

CT Indication Sensitivity Specificity PPV NPV Accuracy

Clinical judgement

% (95% CI) 100

(88.8-100.0) 34.6

(22.2-48.6) 46.3

(41.6-51.1) 100

(82.4-100.0) 58.1

(47.0-68.7)

NICE

% (95% CI) 93.6

(78.6-99.2) 76.4

(62.9-86.8) 69.1

(57.9-78.4) 95.5

(84.5-98.8) 82.6

(72.9-89.9)

Hence with these new results, several corrections have been made in the abstract as well as the manuscript:

Line 44 – “specificity of 93.6% and 76.4% with 81.7% accuracy” has been replaced by “specificity of 93.6% and 76.4% with 82.6 % accuracy”

Line 45 – “34.5, and 54.8 respectively” has been replaced by “34.6%, and 58.1% respectively”

Line 302, “specificity of 93.6% and 76.4% with 81.7%” was replaced with “specificity of 93.6% and 76.4% with 82.6% accuracy in this study”

Once again, I would like to thank the reviewers for their valuable suggestions. Hoping for a positive response.

---

## [Decision Letter · Decision Letter 1]

5 Jul 2021

Impact of implementation of the national institute for health and clinical excellence (NICE) head injury guideline in a tertiary care center emergency department: A pre and post-intervention study.

PONE-D-21-09113R1

Dear Dr. Pradhan,

We’re pleased to inform you that your manuscript has been judged scientifically suitable for publication and will be formally accepted for publication once it meets all outstanding technical requirements.

Kind regards,

Belinda J Gabbe, PhD

Academic Editor

PLOS ONE

Additional Editor Comments (optional):

Reviewers' comments:

Reviewer's Responses to Questions

**Comments to the Author**

1. If the authors have adequately addressed your comments raised in a previous round of review and you feel that this manuscript is now acceptable for publication, you may indicate that here to bypass the “Comments to the Author” section, enter your conflict of interest statement in the “Confidential to Editor” section, and submit your "Accept" recommendation.

Reviewer #1: All comments have been addressed

Reviewer #2: All comments have been addressed

2. Is the manuscript technically sound, and do the data support the conclusions?

Reviewer #1: Yes

Reviewer #2: Yes

3. Has the statistical analysis been performed appropriately and rigorously? 

Reviewer #1: Yes

Reviewer #2: Yes

4. Have the authors made all data underlying the findings in their manuscript fully available?

Reviewer #1: Yes

Reviewer #2: Yes

5. Is the manuscript presented in an intelligible fashion and written in standard English?

Reviewer #1: Yes

Reviewer #2: Yes

6. Review Comments to the Author

Reviewer #1: I would like to thank the authors for their detailed answers and the revision of the manuscript after my comments.

Reviewer #2: Thank you for revising the paper. I enjoyed reading it and it is now suitable for publication in my opinion

Y

7. PLOS authors have the option to publish the peer review history of their article (what does this mean?). If published, this will include your full peer review and any attached files.

Reviewer #1: **Yes: **Omar Bouamra

Reviewer #2: No

---

## [Editor Report · Acceptance letter]

8 Jul 2021

PONE-D-21-09113R1 

Impact of implementation of the national institute for health and clinical excellence (NICE) head injury guideline in a tertiary care center emergency department: A pre and post-intervention study. 

Dear Dr. Pradhan:

I'm pleased to inform you that your manuscript has been deemed suitable for publication in PLOS ONE. Congratulations! Your manuscript is now with our production department. 

Kind regards, 

on behalf of

Professor Belinda J Gabbe 

Academic Editor

PLOS ONE